# A Comprehensive Pan-Cancer Analysis of the Regulation and Prognostic Effect of Coat Complex Subunit Zeta 1

**DOI:** 10.3390/genes14040889

**Published:** 2023-04-10

**Authors:** Ye Hong, Zengfei Xia, Yuting Sun, Yingxia Lan, Tian Di, Jing Yang, Jian Sun, Miaozhen Qiu, Qiuyun Luo, Dajun Yang

**Affiliations:** 1State Key Laboratory of Oncology in South China, Collaborative Innovation Center for Cancer Medicine, Guangzhou 510060, Chinaqiumzh@sysucc.org.cn (M.Q.); 2Department of Pediatric Oncology, Sun Yat-Sen University Cancer Center, Guangzhou 510060, China; 3Department of Experimental Research, Sun Yat-Sen University Cancer Center, Guangzhou 510060, China; 4Department of Medical Oncology, Sun Yat-Sen University Cancer Center, Guangzhou 510060, China; 5Department of Clinical Research, The Third Affiliated Hospital of Sun Yat-Sen University, Guangzhou 510060, China; sjian@mail.sysu.edu.cn; 6Department of Cancer Research, The Eighth Affiliated Hospital of Sun Yat-Sen University, Shenzhen 518033, China

**Keywords:** COPZ1, pan-cancer analysis, stemness signature, prognosis, tumor microenvironment

## Abstract

The Coatomer protein complex Zeta 1 (COPZ1) has been reported to play an essential role in maintaining the survival of some types of tumors. In this study, we sought to explore the molecular characteristics of COPZ1 and its clinical prognostic value through a pan-cancers bioinformatic analysis. We found that COPZ1 was extremely prevalent in a variety of cancer types, and high expression of COPZ1 was linked to poor overall survival in many cancers, while low expression in LAML and PADC was correlated with tumorigenesis. Besides, the CRISPR Achilles’ knockout analysis revealed that COPZ1 was vital for many tumor cells’ survival. We further demonstrated that the high expression level of COPZ1 in tumors was regulated in multi-aspects, including abnormal CNV, DNA-methylation, transcription factor and microRNAs. As for the functional exploration of COPZ1, we found a positive relationship between COPZ1’s expression and stemness and hypoxia signature, especially the contribution of COPZ1 on EMT ability in SARC. GSEA analysis revealed that COPZ1 was associated with many immune response pathways. Further investigation demonstrated that COPZ expression was negatively correlated with immune score and stromal score, and low expression of COPZ1 has been associated to more antitumor immune cell infiltration and pro-inflammatory cytokines. The further analysis of COPZ1 expression and anti-inflammatory M2 cells showed a consistent result. Finally, we verified the expression of COPZ1 in HCC cells, and proved its ability of sustaining tumor growth and invasion with biological experiments. Our study provides a multi-dimensional pan-cancer analysis of COPZ and demonstrates that COPZ1 can serve as both a prospective target for the treatment of cancer and a prognostic marker for a variety of cancer types.

## 1. Introduction

At present, one of the major factors endangering human health is cancer, and the number of people dying from cancer is increasing gradually every year [1]. Proto-oncogene mutations or amplifications, combined with the deactivation of tumor suppressor genes, enable cancer cells to proliferate indefinitely and to resist cell death [2,3]. Targeted inhibitors for HER2, EGFR, and BRAF, and immune checkpoint blockade have achieved certain efficacy in the treatment of tumors, but the occurrence of drug resistance is an important reason to limit their efficacy [4,5,6,7]. Therefore, The search for novel cancer treatment targets and biomarkers is a top priority. COPZ1 is a component of coatomer protein complex I and is involved in the maturation of endosomes, autophagy, and the construction of coated vesicles on the Golgi membrane [8,9,10,11]. COPZ2 is a similar complex that also belongs to the coatomer protein complex family. The most difference between them is their expression pattern: COPZ1 expresses in most cancer cells and normal cells, while COPZ2 expresses in the normal cells and is down-regulated in cancer cells [12]. Previous research has demonstrated that COPZ1 is indispensable for many types of cancer since the cancer cell possessed a low expression of the isoform COPZ2, including breast cancer, ovarian cancer, and prostate cancer [12]. Depletion of COPZ1 can lead to thyroid tumor cell death both in vitro and in vivo [13]. Besides, Silencing COPZ1 causes the endoplasmic reticulum (ER) stress to increase and the type I IFN pathway activated, and then promotes the secretion of a variety of inflammatory molecules and enhances anti-tumor immunity in thyroid cancer [14]. Another research found that knockdown of COPZ1 leads to ferroptosis by inducing NCOA4-mediated autophagy in human GBM cells [15]. These results imply that COPZ1 plays a critical role in maintaining the survival of tu-mor cells and may represent a viable therapeutic target for the treatment of cancer. However, the regulation mechanism and prognostic potential of COPZ1 in different cancers remains unclear. The pan-cancer bioinformatic analysis may therefore enable us to better elucidate its function in human cancer.

In this study, bioinformatics analysis was carried out to comprehensively explored the function and regulation mechanism of COPZ1 in pan-cancer by combining GTEx and TCGA data. We found that COPZ1 could be a robust prognostic marker for various types of cancer. The CNV alteration frequency, DNA methylation, potential regulated transcription factors, and microRNAs of COPZ1 were evaluated. A high level of COPZ1 is associated with increased stemness and hypoxia scores. We further investigated COPZ1’s role in the tumor microenvironment (TME), as well as its relationship to the immunological score, stromal score, immune cell infiltration percentage, cytokines level, immune activator genes expression, and immune inhibitor genes expression. Taken together, Our findings can serve as support for further research into COPZ1’s function in malignancies and the possibility of COPZ1-specific cancer therapies.

## 2. Materials and Methods

### 2.1. Downloading Data, Analyzing Differential Expression and Exploring the Significance of COPZ1 in Tumor

The Cancer Genome Atlas (TCGA) database (http://cancergenome.nih.gov/, accessed on 6 January 2023) and Cancer Cell Line Encyclopedia (CCLE) database (https://sites.broadinstitute.org/ccle/, accessed on 15 January 2023) were used to retrieve the methylation, single nucleotide polymorphism (SNP), copy number variation (CNV), and transcriptome sequencing data. The transcription expression data of individual cancer was downloaded from the UCSC Xena (https://xenabrowser.net/, accessed on 18 January 2023) in count format. We used the TIMER database (http://timer.cistrome.org/, accessed on 18 January 2023) and GEPIA2 database (http://gepia.cancer-pku.cn/, accessed on 15 January 2023) to explore the gene expression level of COPZ1 between the tumor and corresponding normal tissues [16]. In addition, the protein expression of COPZ1 in tumor and normal tissues was compared by using the UALCAN portal (http://ualcan.path.uab.edu/analysis-prot.html, accessed on 1 February 2023). The miRNA data were downloaded from the human genome annotated file in UCSC (http://genome.ucsc.edu/, accessed on 30 January 2023) and extracted the gene names and corresponding gene types were.

GEPIA2 database is accustomed to evaluating the prognostic role of COPZ1 in the overall survival (OS) of cancer patients [17]. The median value of COPZ1 expression was used to divide patients into two groups to analyze survival differences. Significant was defined as a *p* value 0.05. Additionally, we used the downloaded CRISPR data (CRISPR (DepMap 22Q2 Public+Score, Chronos)) to examine how COPZ1 affects the development and survival of tumor cells.

### 2.2. Exploring the Association of COPZ1 Expression and Special Signature

We downloaded the gene set of proliferation and hypoxia signature from published articles and used the single sample Gene set enrichment analysis (ssGSEA) to calculate the hypoxia score of each patient. We also downloaded stemness indices of patients from the published article. Between the COPZ1 high-expression and low-expression groups, the hypoxia score and stemness indices were evaluated. Meanwhile, the tumor mutation burden (TMB) and microsatellite instability of each patient were calculated and compared.

### 2.3. Identifying Differential Expressed Genes and Conducting Functional Enrichment Analysis

We used the “limma” R package for differential gene expression analysis between COPZ1 high and low-expression patients. The differential expressed genes were defined as |log FC| ≥ 1 and FDR < 0.05. The functional annotation of biological process (BP) and Kyoto Encyclopedia of Genes and Genomes (KEGG) Analysis for DEGs between high and low COPZ1 expression groups were conducted with the “clusterProfiler” package of R based on Org. Hs.eg.db comment pack. To further explore the pathway, Gene Set Enrichment Analyses (GSEA) were carried out based on the gene set “c5.go.bp.v7.5.1.entrez.gmt” and “c2.cp.reactome.v7.5.1.entrez.gmt” downloaded from the Molecular Signatures Databases (MSigDB; https://www.gsea-msigdb.org/gsea/msigdb/index.jsp, accessed on 30 January 2023), with the log2(FoldChange) data from the DEGs analysis between high and low COPZ1 expression group. The absolute normalized enrichment scores (NES) >1 and nominal *p*-values <0.05 were used to determine the enrichment levels and statistical significance. R package “GSEABase” and “clusterProfiler” (version 3.10) was used to perform GSEA and create the GSEA plot.

### 2.4. Evaluating the Association between COPZ1 Expression and Tumor Immune Microenvironment

We first compared the expression level of immune activation genes between COPZ1 high- and low-expression patients. Next, we used the CIBERSRT algorithm to deconvolute the infiltration of immune cells in the tumor microenvironment (TME). Additionally, the immunological score and stromal score between COPZ1 high-expression and low-expression patients were assessed using the ESTIMATE algorithm. We also acquired the gene sets of 29 TME signatures and calculated each patient’s score by ssGSEA. Last, we used the published tool “Cytosig” to evaluate the activity of 43 kinds of cytokine pathways activity. To explore the impact between COPZ1 level and M2 invasion in each cancer, the results of M2 infiltration score generated by different methods were downloaded from the TIMER database. The “ggpubr” R package was used to compare and visualize the above characteristics.

### 2.5. Defining the Regulation Mechanism of COPZ1 Expression in Tumor

We analyzed the methylation, SNP, CNV and transcriptome sequencing data to define whether epigenetics, genomics, transcription factors (TFs) and microRNA caused the various ways that tumors and healthy tissues express COPZ1. The potential TFs of COPZ1 were predicted and downloaded from the Database of Human Transcription Factor Targets (hTFtarget) (http://bioinfo.life.hust.edu.cn/hTFtarget#!/, accessed on 24 January 2023).

### 2.6. Cell Lines and Reagents

Human normal human liver cell LO2 and HCC cell lines were obtained from the American Type Culture Collection (ATCC, Manassas, VA, USA). Mycoplasma detection and short tandem repeat (STR) analysis allowed for the identification of all the cell lines. The aforementioned cells were grown in DMEM (Gibco, Billings, MT, USA) supplemented with 10% fetal bovine serum (FBS), 1% antibiotics, and 37 °C in a humidified environment with 5% CO_2_.

### 2.7. RNA Extraction and qRT-PCR Analysis

Total RNA from HCC cell lines was extracted using RNA Quick Purification kit (ESscience Biotech, Shanghai, China). RNA was reverse transcribed into cDNA with the RT kit (ESscience Biotech, Shanghai, China). Real-time PCR was performed with ChamQTM SYBR^®^ qPCR Master Mix kit (Vazyme Biotech, Nanjing, China). The mRNA expression levels of the genes were calculated in relation to that of -Actin for each sample, which was done in triplicate. The primers used in this study are listed as follows:COPZ1-F: 5′-GATGGAGATCGACTTTTTGCCA-3′COPZ1-R: 5′-TCAGTCCGATGGGTCTTGTTG-3′GAPDH-F: 5′-GGAGCGAGATCCCTCCAAAAT-3′GAPDH-R: 5′-GGCTGTTGTCATACTTCTCATGG-3′

### 2.8. Western Blotting

PLC/PRF/5 and Huh7 cells were collected, washed twice with pre-cooled PBS, and then lysed for 30 min on ice using a cell lysis buffer (Beyotime, P0013, Haimen, China), containing 1 percent protease inhibitor (PMSF), and 1 percent phosphokinase inhibitor. The supernatant protein lysate was then collected. Protein concentration was measured by a Pierce BCA Protein Assay kit (Thermo Scientific, Waltham, MA, USA). A PVDF membrane was then used to transfer the separated cellular lysates that had been separated using a 10% SDS-PAGE gel. The chosen primary antibodies were probed with the PVDF membrane overnight at 4 °C after it had been masked with 5% BSA buffer for an hour at room temperature. The primary antibodies against COPZ1 (1:1000; Proteintech, 20440-1-AP, Sankt Leon-Rot, Germany) and GAPDH (1:5000; AM102013) were used. After that, 1×TBST (washing buffer) washed the mem-brane 3 times for 10 min each time. After being incubated with a secondary antibody for an hour at room temperature, the protein membrane was washed with 1×TBST three times for a total of ten minutes. The secondary anti-mouse (sc-2005) and anti-rabbit (sc-2004) antibodies were purchased from Santa Cruz Biotechnology. An ECL chemiluminescence hypersensitive colorimetric kit and a chemiluminescent imaging system were used for signal generation and detection.

### 2.9. siRNA Transfection

The transfection regent and siRNA for COPZ1 and negative control were purchased from GenePharma Group Inc (Suzhou, China). The PLC/PRF/5 and Huh7 cells were seeded in a 6-well plate with 5 × 10^5^ cells per well. According to the manufacturer’s instructions, the cells were incubated for another 24 h after being transfected with the siRNA using GP-transfect-Mate regent and the culture medium was replaced 8 h later. After that, the cells were collected for proliferation and migration ability detection.

### 2.10. Cell Proliferation Detection

The cell counting kit 8 (CCK8) (Dojindo, Kumamoto, Japan) and colony formation assay were applied to measure the cell proliferation of HCC cells transfected with COPZ1 siRNA. For the CCK8 assay, PLC/PRF/5 and Huh7 cells were plated in a 96-well culture plate, with 2 × 10^3^ cells per well. Each well received 100 μL of new media containing 10% CCK8 at the predetermined periods, and each well was then incubated for 2 h at 37 °C. The absorbance was measured under the wavelength of 450 nm. For colony formation assay, PLC/PRF/5 and Huh7 cells were seeded in a 6-well plate, with 2000 cells per well, and replaced the culture medium every 3–4 days. After two weeks, the cells were stained with 0.5% crystal violet dissolved in methanol for 30 min at room temperature. Crystal violet dye was cleaned with running water and the number of colonies was calculated.

### 2.11. Cell Migration and Invasion Assay

PLC/PRF/5 and Huh7 cells (2 × 105 cells) were seeded on the upper chambers of the transwell chamber (Pore size 8.0 μm, Corning, NY, USA). Serum-free medium was placed in the upper chambers, and 750 uL of 50% FBS culture medium was placed in the lower chambers. The chamber was removed after 36 h, and the interior membrane was cleaned with a cotton swab to remove any adhering cells, before being stained with 0.5% crystal violet dissolved in methanol for 30 min. Then, the chambers were then allowed to dry at room temperature before being imaged using a microscope. Using a microscope, the migratory cells were counted in five selected at random fields each chamber. Three independent runs of each experiment were made.

### 2.12. Statistical Analysis

Pearson correlation analysis was used to explore the correlations between variables. T-test or Wilcoxon test was used to compare continuous variables between binary groups. For comparisons of more than two groups, Analysis of variance or Kruskal-Wallis tests were used to compare the differences. The Chi-square test was used to compare the difference in category variables. R 4.2.3 was used to implement all statistical studies (https://www.r-project.org, accessed on the 16 March 2023). *p* values were symmetrical. Statistics were considered significant with *p* values under 0.05. Other methods are available in Appendix A.

## 3. Results

### 3.1. COPZ1 Expression Profile in Pan-Cancers and Corresponding Normal Tissues

The expression profile of COPZ1 was analyzed in pan-cancer and corresponding normal tissue types through TIMER2 and GEPIA2. In the TIMER2 database, the RNA expression level of COPZ1 in TGCA data showed that COPZ1 was overexpressed in multiple types of cancers compared to the corresponding normal tissues, including bladder urothelial carcinoma (BLCA), breast carcinoma (BRCA), cholangiocarcinoma (CHOL), colon adenocarcinoma (COAD), esophageal carcinoma (ESCA), head and neck squamous cell cancer (HNSC), kidney renal papillary cell carcinoma (KIRP), liver hepatocellular cancer (LIHC), lung adenocarcinoma (LUAD), lung squamous cell cancer (LUSC), prostate adenocarcinoma (PRAD), rectum adenocarcinoma (READ), stomach adenocarcinoma (STAD) and uterine corpus endometrial carcinoma (UCEC) (Figure 1A). However, we found that COPZ1 with a lower expression level in kidney chromophobe (KICH) and thyroid carcinoma (THCA) compared with normal tissues. Besides, for those cancers without corresponding normal tissues in TIMER2, including lymphoid neoplasm diffuse large B-cell lymphoma (DLBC), glioblastoma multiforme (GBM), thymoma (THYM), acute myeloid leukemia (LAML), ovarian serous cystadenocarcinoma (OV), testicular germ cell tumor (TGCT), brain lower grade glioma (LGG), and skin cutaneous melanoma (SKCM), we analyzed the expression of COPZ1 via GEPIA2 dataset. The results demonstrated that tumor tissue had much greater levels of COPZ1 expression than normal tissue in DLBC, GBM, THYM, OV, TGCT, LGG and SKCM, but lower in tumor tissue in LAML (Figure 1B). The COPZ1 expression in the rest cancers was also explored in the GEPIA database with GTEx normal data, and similar trends occurred in TIMER also observed in the GEPIA. Cancers with statistical significance include the BRCA, CHOL, COAD, READ, LIHC and STAD (Appendix A). We further assessed the differential protein expression of COPZ1 between tumor and normal tissue in the Clinical Proteomic Tumor Analysis Consortium (CPTAC) dataset. Only ten types of cancer protein expression data were available in this database. In tumor tissue compared to healthy tissue, the amount of COPZ1 protein expression was noticeably increased in breast cancer, colon cancer, OV, UCEC, lung cancer, HNSC and liver cancer. (Figure 1C). In pancreatic cancer, the albumin level of COPZ1 was lower in tumor tissue when compared to normal tissue.

### 3.2. High COPZ1 Expression Predicts Poor Prognosis in Multiple Cancers

We next investigated the correlation between COPZ1 expression and survival outcomes in different tumors. As shown in Figure 2A, high expression of COPZ1 was associated with poor overall survival in BLCA (HR =1.7, *p* = 0.00057), BRCA (HR =1.4, *p* = 0.03), HNSC (HR = 1.6, *p* = 0.0014), LIHC (HR = 1.5, *p* = 0.028), and LUAD (HR = 1.4, *p* = 0.018), while low expression of the COPZ1 was linked to the poor OS in UCEC (HR = 0.41, *p* = 0.017). With comprehensive consideration of the results of COPZ1 differential expression in cancer and corresponding normal tissue, and the association between COPZ1 expression and patients’ survival, we speculated whether COPZ1 was essential for those cancer cells’ survival. Next, we analyzed the data for CRISPR-modulated COPZ1 knockout by Achilles’ project. We discovered that knockout of COPZ1 could significantly inhibit tumor cell proliferation in the majority of cancer cell lines derived from BLCA, BRCA, HNSC, LIHC and LUAC (Figure 2B). Therefore, the above six types of cancer were selected for further investigation of the function and regulation mechanism of COPZ1.

We next investigated the correlation between COPZ1 expression and survival outcomes in different tumors. As shown in Figure 2A, high expression of COPZ1 was associated with poor overall survival in BLCA (HR = 1.7, *p* = 0.00057), BRCA (HR = 1.4, *p* = 0.03), HNSC (HR = 1.6, *p* = 0.0014), LIHC (HR = 1.5, *p* = 0.028), and LUAD (HR = 1.4, *p* = 0.018), while low expression of the COPZ1 was linked to the poor OS in UCEC (HR = 0.41, *p* = 0.017). With comprehensive consideration of the results of COPZ1 differential expression in cancer and corresponding normal tissue, and the association between COPZ1 expression and patients’ survival, we speculated whether COPZ1 was essential for those cancer cells’ survival. Next, we analyzed the data for CRISPR-modulated COPZ1 knockout by Achilles’ project. We discovered that knockout of COPZ1 could significantly inhibit tumor cell proliferation in the majority of cancer cell lines derived from BLCA, BRCA, HNSC, LIHC and LUAC (Figure 2B). Therefore, the above six types of cancer were selected for further investigation of the function and regulation mechanism of COPZ1.

### 3.3. Investigation of the Regulation Mechanism of the COPZ1 Expression

To investigate the potential regulatory mechanism that resulted in the differential expression level of COPZ1 between normal and tumor tissues, we firstly analyzed the alteration frequency of COPZ1 in the above six kinds of cancer in the cBioportal database. According to the findings, endometrial cancer had the highest frequency of COPZ1 alterations, the majority of which were amplification mutations (Figure 3A). However, we found no significant correlation between COPZ1 expression level and COPZ1 mutation among six types of cancer. Furthermore, the CNV level of COPZ1 in the tumor tissue and the comparable normal tissue differs noticeably. Through the analysis of the correlation between COPZ1 expression and the CNV of COPZ1, we found that CNV is one of the reasons that lead to COPZ1 abnormal expression (Appendix A).

Next, we performed a methylation analysis of the COPZ1 promoter region, and we found that there were differences in the methylation of the COPZ1 promoter region between normal and tumor tissues. We subsequently assessed the association between the representation of the COPZ1 gene and those differential DNA-methylated CpG sites in six types of cancer (Figure 3B–G). The findings showed that COPZ1 expression negatively correlated with CpG site cg17271146 in LIHC (R = −0.25, *p* = 1.3 × 10^−5^) and LUAD (R = −0.36, *p* = 1.8 × 10^−15^), CpG site cg03948415 in BRCA (R = −0.20, *p* = 2.8 × 10^−8^), CpG site cg00576388 (R = −0.18, *p* = 0.00024) and CpG site cg11910118 (R = −0.20, *p* = 4.1 × 10^−5^) in UCEC. There was a positive relation between COPZ1 gene expression with CpG site cg17271146 in BRCA (R = 0.18, *p* = 7.9 × 10^−7^).

In addition to exploring and analyzing the genomic and epigenetic regulation mechanism of COPZ1 gene expression, we also analyzed the correlation between COPZ1 expression level and its potential regulated transcription factors. One hundred and seventeen potential transcription factors of COPZ1 were downloaded from the hTFtarget database. We firstly performed differential expression analysis of COPZ1 transcription factors between normal and tumor tissues. Considering that the TFs which promote the expression of COPZ1 should exert the same expression pattern as COPZ1, we selected the TFs which were upregulated in tumor tissues for further analysis. We next carried out the correlation analysis between the COPZ1 expression and the expression of upregulated TFs among the six kinds of cancer. In BLCA, the top three TFs positively linked to COPZ1 expression were SMARCB1 (R = 0.28), CBX8 (R = 0.21) and CEBPA (R = 0.19), while the PML was the most negative TF correlated with COPZ1 expression (R = −0.13) (Appendix A). GATA3 (R = 0.50), CDK7 (R = 0.44), MYB (R = 0.34) and ZNF92 (R = 0.25) were the TFs most strongly correlated with COPZ1 expression in BRCA, while TCF3 (R = −0.33), SPI1 (R = −0.25) and PRAME (R = −0.25) were the most inversely connected to COPZ1 level (Appendix A). SUMO (R = 0.35), E2F1 (R = 0.33) and USF1 (R = 0.30) were the top three TFs that had a positive correlation with COPZ1 expression, while KDM5B (R = −0.25), TFAP2A (R = −0.24) and ETS1 (R = −0.23) were most negatively associated with COPZ1 expression in HNSC (Appendix A). In LIHC, there were many TFs significantly correlated with COPZ1 expression and most of them were positively correlated with COPZ1 expression, and the correlation value ranked top five were TCF3 (R = 0.55), CBX3 (R = 0.54), CDK7 (R = 0.52), SUMO2 (R = 0.52) and USF1 (Appendix A). In LUAD, CDK7 (R = 0.30), MAZ (R = 0.25), PRAME (R = 0.22) and CBX3 (R = 0.20) were the most positive associated TFs while TCF3 (R = −0.14) was the negative associated TF for COPZ1 (Appendix A). In UCEC, CDK7 (R = 0.30), FOXA2 (R = 0.24), MYB (R = 0.23) and CBX8 (R = 0.20) are TFs that were positively correlated with COPZ1 expression, while ARID3A (R = −0.15) was a negative TF for COPZ1 (Appendix A).

MicroRNA can mediate cleavage and subsequent degradation of mRNA, which is one of the mechanisms that regulate gene expression [18]. Therefore, we wanted to investigate the MicroRNAs which may regulate the COPZ1 expression. We analyzed the differentially expressed MicroRNAs between normal and tumor tissues, and then screened out the MicroRNAs that were significantly downregulated in tumor tissues. Next, we explored the correlation between the levels of these significantly downregulated microRNAs and COPZ1 expression. In BLCA, COPZ1 expression had a strong inverse relationship with MIR 23A, MIR24-2, MIR27A and MIR568 (Appendix A). In BRCA, the expression of COPZ1 was significantly negatively associated with MIR 186, MIR221, MIR23B, MIR27A, MIR27B, MIR568 and MIR590 (Appendix A). MIR221, MIR23A, MIR30C2, MIR3677, MIR568 and MIRLET7D were negatively correlated with COPZ1 expression in LUAD (Appendix A). In UCEC, COPZ1 showed a negative association with MIR221, MIR23A, MIR23B, MIR27B and MIR4786 (Appendix A). However, in HNSC and LIHC, only MIR568 and MIR621 showed a correlation with COPZ1 expression, respectively (Appendix A).

### 3.4. COPZ1 Positively Correlated with Stemness Score and Hypoxia Score

Since stemness and hypoxia are two important features of tumors, we next investigated how the tumor’s stemness and hypoxia index were related to COPZ1 expression. With the mRNAsi (mRNA expression-based stemness index) of six types of cancers acquired from a published article [19], in both of the six different types of tumors, we observed that COPZ1 was highly associated with the mRNAsi score (Figure 4A). We then explored the stemness score between the COPZ1 high and low expression groups. The results indicated that in comparison to the low-expression group, the stemness score in the COPZ1 high-expression group was significantly higher (Figure 4B). For hypoxia analysis, as shown in Figure 4C, the expression of COPZ1 positively correlated with the hypoxia score in BLCA (R = 0.14, *p* = 0.004), HNSC (R = 0.21, *p* = 1.4 × 10^−6^), LUAD (R = 0.44, *p* < 2.2 × 10^−16^) and UCEC (R = 0.23, *p* = 4.3 × 10^−8^). Besides, in BLCA, BRCA, HNSC, and LUAD, the hypoxia score was noticeably greater in the group with high COPZ1 expression. (Figure 4D). These results demonstrated that COPZ1 might aid in the development of the tumor, but more research is required to determine the precise mechanism.

### 3.5. The Relationship between COPZ1 Expression and Tumor Metastasis

Metastasis is a critical and death-leading feature of tumor. Based on the fact that COPZ1 has a high expression in tumor tissue, it is worth to further exploring the relationship of COPZ1 with the metastasis in pan-cancer. We firstly scanned the correlation coefficient of COZP1 and EMT biomarkers, including the pro-metastasis markers, CDH2 (N-cad), SNAI1 (Snail), SNAI2 (Slug) and VIM (Vimentin), and inhibitory marker, CDH1 (E-cad). Our heatmap showed that in most cancers, the pro-metastasis and inhibitory markers were both positively or negatively correlated with the COPZ1 expression (Figure 5A), which may suggest the vague role of COPZ1 in the tumor metastasis. However, in SARC, a significant negative correlation with the CDH1 and a positive correlation with the pro-metastasis markers were observed (Figure 5A). Thus, we divided the TCGA SARC cohort into high and low COPZ1 expression groups by median value, and the DEGs were gained (Figure 5B). The REACTOME enrichment analysis showed that the DEGs mainly participated in the ECM proteoglycans and integrin cell surface interactions (Figure 5C). What’s more, the possible mechanisms were examined using the GSEA analysis. The BP and HALLMARK enrichment showed that EPITHELIAL TO MESENCHYMAL TRANSITION was activated in the high ZOPC1 expression group (Figure 5D,E). The KEGG and REACTOME signal pathways analysis showed that WNT SIGNALING PATHWAY and SIGNALING BY INTERLEUKINS were activated in the high ZOPC1 expression group (Figure 5F,F). Therefore, higher ZOPC1 expression may relate to the activation of metastasis progress.

### 3.6. Correlation Analysis of COPZ1 Expression with Tumor Mutation Burden and Microsatellite Instability

TMB and MSI are important biomarkers for tumors. Hence, sought to investigate the relationship between the TMB and MSI levels and COPZ1 ex-pression. The analysis revealed that no significant association was found between COPZ1 expression and TMB in these six cancer types, but TMB level was higher in the COPZ1 high expression group than in the COPZ1 low expression group in LIHC (Appendix A). COPZ1 expression was positively correlated with the MSI in HNSC (R = 0.12, *p* = 0.0093) and UCEC (R = 0.17, *p* = 0.00011) (Appendix A). In BRCA and LUAD, MSI levels were higher in the group with low COPZ1 expression than in the group with high COPZ1 expression (Appendix A). Taken together, this result showed that the correlation of COPZ1 with TMB and MSI was not significant.

### 3.7. Pathway Enrichment Based on the Expression of COPZ1

To explore the biological function that COPZ1 may involve in, we carried out the KEGG pathway GSEA analysis. The most correlated pathways with the expression level of COPZ1 among the six types of cancer were shown in Figure 6 and Appendix A. We discovered a favorable correlation between high COPZ1 expression and oxidative phosphorylation, cell cycle, DNA replication, mismatch repair and platinum drug resistance (Appendix A). However, COPZ1 low expression was negatively correlated with multiple immune-related pathways, including TNF signaling pathway, Chemokine signaling pathway, T cell receptor signaling pathway, Toll-like receptor signaling pathway, and NOD-like receptor signaling pathways (Figure 6). These results indicated a correlation of COPZ1 with tumor immunity.

### 3.8. Correlation Analysis of COPZ1 Expression with Immune Cells and Tumor Microenvironment

Since the pathway enrichment results indicated that COPZ1 was correlated with many immune response pathways, we wanted to find out how COPZ1 expression related to the tumor microenvironment and immunomodulator expression. We used the “estimate” R package to evaluate the immune score and stromal score between COPZ1 high-expressed and low-expressed groups. The results showed that the COPZ1 low expressed group had a higher immune score in BRCA, HNSC, and LUAD, but not in BLCA, LIHC and UCEC (Figure 7A–F). Additionally, in both of the six cancer types, reduced expression of COPZ1 was substantially linked to a higher stromal score (Figure 7G–L). We next explored the association between the COPZ1 expression and infiltration of immune cells by the CIBERSORT algorithm. As presented in Figure 8A–F, except for the BLCA, in the other five kinds of cancer, low expression of COPZ1 exhibited more antitumor immune cells infiltration, including B cells, neutrophils, CD8+T cells, CD4+T cells and M1 macrophages, while high expression of COPZ1 associated with more Tregs infiltration. We further performed ssGSEA to calculate the score of 29 types of gene signatures of TME in each patient and compared these scores in COPZ1 high and low expression patients. The findings revealed that the COPZ1 low expression group had many signature scores that were elevated (Figure 8G–L), such as Effector_cells, Antitumor_cytokines, M1_signature, etc. These findings suggested that the modulation of the tumor immune microenvironment may involve COPZ1.

We then assessed the correlation between the COPZ1 expression and the immunomodulator expression. We found that COPZ1 exhibited a negative association with most immune activator genes, including CD28, CXCL12, CD40, IL6R, TNFSF18 and CD27 (Appendix A). However, in terms of immunosuppressive genes, low expression of COPZ1 also correlated with a higher level of immune inhibitors, including TGFB1, CTLA4, PDCD1 and TIGIT (Appendix A). We finally analyzed the level of 43 kinds of cytokines’ pathway activity between high and low COPZ1 expression groups. As shown in Appendix A, COPZ1 expression was negatively associated with many cytokines’ pathway activity, including CXCL12, GCSF, IFN-G, and IL12. These results demonstrated that COPZ1 has a complicated and varied role in tumor immunity.

### 3.9. Correlation Analysis of COPZ1 Expression with Hypoxia and M2 Infiltration Level

As mentioned above, the higher expression of COPZ1, the higher hypoxia score was observed. To investigate the connection between hypoxia and COZP1, we analyzed the correlation of COPZ1 and HIF1A as well as the LOXL2 in 33 cancer types. The result showed that except for the glioblastoma multiforme and esophageal carcinoma, COPZ1 had a positive correlation with the HIF1A in most of the rest cancer (Figure 9A). As for the LOXL2, another important downstream target of the HIF1A, the similar results were observed (Figure 9B). Then, we further analyze the correlation between the COPZ1 expression and the M2 infiltration. The COPZ1 expression had a positive correlation with M2 infiltration score from the CIBERSORT-ABS and CIBERSORT (Figure 9C). What’s more, the COPZ1 also showed a positive correlation with the M2 markers (Figure 9D). These results demonstrated that COPZ1 had a close relation with the hypoxia and the M2 infiltration, which may mediate the immunosuppressive microenvironment.

### 3.10. High COPZ1 Levels Contribute to the Proliferation and Migration of HCC

Next, to better verify the function of COPZ1 in tumorigenesis, HCC was selected for further experimental evaluation with the rich study in the HCC. We found that the mRNA level of COPZ1 was significantly higher in HCC tissues than the adjacent non-tumor tissue both in GSE17856 and GSE14520 datasets, which was similar to the result generated by TCGA dataset (Figure 10A,B). Western blot assay showed that the protein expression of COPZ1 was higher in most HCC cell lines than the normal human liver cell LO2 (Figure 10C). To investigate the function of COPZ1 in HCC, we silenced COPZ1 using siRNA in PLC/PRF/5 and Huh7 cells. The mRNA and protein levels of COPZ1 in PLC/PRF/5 and Huh7 cells were dramatically reduced after COPZ1 siRNA transfection (Figure 10D). And then, the results of CCK-8 assay and colony formation assay confirmed that the cell proliferation ability of PLC/PRF/5 and Huh7 cells was suppressed after knockdown of COPZ1 (Figure 10E–G). Besides, the transwell test showed that PLC/PRF/5 and Huh7 cell migration was decreased by COPZ1 knockdown (Figure 10H,I). Taken together, these findings suggest that COPZ1 maintains the proliferation and migration ability of HCC cells.

## 4. Discussion

In this study, we carried out a bioinformatic analysis to explore the expression profile, molecular characteristics and clinical prognosis value of COPZ1 in pan-cancers. Compared to the normal tissues, we discovered that numerous cancer types have significant levels of COPZ1 mRNA expression, and the COPZ1 protein expression level was considerably greater in breast cancer, colon cancer, OV, UCEC, lung cancer, HNSC and liver cancer tissues. Furthermore, Our results revealed that a high level of COPZ1 representation was related with a poor prognosis in BLCA, BRCA, HNSC, LIHC and LUAD. Several studies have reported that COPZ1 is vital for sustaining cancer cell survival in thyroid tumors, breast cancer, ovarian cancer, and prostate cancer [12,13]. Herein, the data from CRISPR Achilles’ project in CCLE confirmed that knockout of COPZ1 could suppress tumor cell viability in most cancer cell lines. Additionally, BMI1 can promote breast cancer cell proliferation and inhibit autophagy by activating COPZ1 transcription [20]. Recent research has discovered an antibody conjugated polymer nanogel system called Nano-ERASER, which could degrade COPZ1 in cancer cells and finally cause cancer cell death [21]. However, the COPZ1 expression was low in tumor tissue in the LAML and PADC. Interestingly, the low COPZ1 expression in these tumors did not reduce their ability of tumorigenesis (Appendix A). The BP and REACTOME analysis of the DEGs showed that they participate in the cell adhesion and EMT related pathways (Appendix A). The GSEA analysis further showed that IFN-r, KRAS signaling, IL-2/STAT5 signaling and TGF-β signaling were activated in low COPZ1 expression group (Appendix A).

To further explore the DEGs and pathways that specific for LAML or PAAD but not for other cancers which might responsible of the negative trend of COPZ1. We found that the DEGs between high and low COPZ1 expression in the LAML and PADC had merely several overlapped genes, but their showed no special expression of LAML or PADC (Appendix A). Then, the DEGs between tumor and normal across 11 types were analyzed and the genes upregulated in LAML or PADC but downregulated in the other tumor, or upregulated in LAML or PADC but downregulated in the other tumor were collected (Appendix A). Unfortunately, no genes were gained for the PADC, the mechanism of low expression level of COPZ1 in PADC might be at the protein level, for the expression level difference of COPZ1 was only observed in the CPTAC database. For the special DEGs for LAML, 32 genes had significantly relation with the COPZ1 (Appendix A). The BP, MF and REATOME analysis showed that those genes mainly participate in the nuclear division, kinase activity and cell cycle (Appendix A). And the GSEA showed the special pathways suppressed in the LAML while activated in the rest 9 tumors, and they mainly participated in the cell cycle (Appendix A). However, those pathways seem not enough to explain the mechanism of low COPZ1 expression in the LAML. To find out the detail pathways regulated the COPZ1 expression in the LAML, more biological experiments are warranted. These results reflected the complicated expression model in different cancer types, but both results highlight the role of COPZ1 in sustaining tumor growth and proliferation and suggest that COPZ1 may be a novel target for cancer therapy, and our discovery may provide a basic understanding for choosing the potential patients who might benefit from the drugs targeted the COPZ1.

To look into possible regulatory mechanisms of high COPZ1 expression in cancer, we carried out analysis from many aspects, including epigenetics, genomics, transcription factors and microRNA. Although COPZ1’s gene alteration frequency is not high, the level of CpG methylation at certain COPZ1 DNA methylation sites were found to be inversely correlated with the gene’s expression. Besides, we also identified many transcription regulators and microRNAs that were positively or negatively correlated with COPZ1 expression in different cancers. Our findings indicate that the expression of COPZ1 is regulated in multi-dimension. However, relevant exploration of post-transcriptional regulation of COPZ1 was warranted in future study.

Tumor stem cells are one of the important causes of tumor recurrence and drug resistance [22,23]. The relationship between COPZ1 and tumor stemness has not been investigated before. In our work, we identified a favorable connection between the stemness score and COPZ1 expression. The stemness score was significantly higher in COPZ1 high-expression group. The stemness score was developed by Sokolov et al., and higher stemness score values was associated with activation of cancer stem cells’ biological processes [24]. Previous research demonstrated that the stemness score was related to immune microenvironment content and PD-L1 levels, and the stemness score was elevated in metastatic tumors [25,26]. Cancer stemness was identified as a prognostic marker for many cancers, and cancer stemness may affect TME immune cell infiltration and immunotherapy efficacy [27,28,29]. The correlation between COPZ1 and stem cancer may be one of the reasons why COPZ1 can maintain the survival of tumor cells.

Due to the rapid and uncontrolled proliferation of tumors, oxygen supply to tumor cells is limited. Hypoxia is one of the common features of almost all solid tumors [30]. Hypoxia can promote tumor plasticity and heterogeneity, and promote a more aggressive and metastatic phenotype, which contributes to tumor progression and therapy resistance [30,31,32,33]. In addition, hypoxia stress can cause immunosuppression in tumor [34,35,36]. In lung cancer, a study reported that HIF1A could mediate the immunosuppressive by the HIF1A/LOXL2/EMT pathways, and our results also showed that COPZ1 had a positive correlation with HIF1A, LOXL2 and EMT [37]. Similarly, more tumor-associated macrophages (TAMs) could be found in the hypoxia area, which would be converted into M2 macrophages with more immunosuppressive activities [38]. Besides, hypoxia can also regulate the function of MDSCs and upregulate the expression of immune checkpoints like PD-L1 [39,40]. In our research, the amount of COPZ1 was favorably correlated with the hypoxia score in many malignancies. The further analysis showed that COPZ1 expression had a positive correlation with the M2 infiltration in most cancer types. These results revealed an immune suppressive role of the COPZ1 in cancers. The relationship between COPZ1 and hypoxia has not been previously reported. Whether hypoxia leads to COPZ1 overexpression or COPZ1 overexpression the occurrence of hypoxia deserves further exploration. In addition, concerning the close relationship between hypoxia and the tumor immune microenvironment, the role of COPZ1 in tumor immune micro-environment needs further investigation.

One of the characteristics of cancer is immune reprogramming [41]. Immune checkpoint inhibitors are promising cancer therapies in recent years, but their clinical efficacy remains limited [42,43]. Therefore, exploring the molecular and regulatory mechanisms related to tumor immunity is beneficial for increasing the effectiveness of cancer immunotherapy and predict the prognosis of patients. An earlier investigation discovered that COPZ1 knockdown might boost the release of several pro-inflammatory cytokines, which would then accelerate dendritic cell maturation and heighten the cytotoxic T cell response [14]. Based on our KEGG pathway GSEA analysis, we found a close correlation between COPZ1 and the tumor immune response pathways, such as TNF signaling pathway, Chemokine signaling pathway, T cell receptor signaling pathway, Toll-like receptor signaling pathway, NOD-like receptor signaling pathways, which indicated that COPZ1 may involve in the tumor immunity. TME is comprised of a variety of immune cells, cytokines, endothelial cells and fibroblasts, which have been identified to influence response to immune therapy [44]. In our current work, we found that there was a clear link between the hypoxia score and the amount of COPZ1 in several cancers. Previous studies revealed that a higher immune score was correlated with better survival outcomes [45,46]. We further discovered that the infiltration of antitumor immune cells was higher in the patients with low COPZ1 expression, such as B cells, Neutrophil, CD8+T cells, CD4+T cells and M1 macrophages. However, high expression of COPZ1 is associated with more Tregs infiltration. Tumor hypoxia is considered to act as a suppressor which limited the immune response via stimulating the synthesis of metabolites and immunosuppressive molecule expression in TME [47]. Hypoxia-induced the up-regulation of HIF-1αcan suppress the migratory activity of effector T cells [48]. The lack of T cell infiltration limited the T cells’ response to immunotherapy [49]. Recent studies revealed that a higher stemness score was closely related to lower expressed PD-L1 and a reduced immune cells infiltration, which may mediate tumor immune evasion [25,50]. Concerning our above findings that COPZ1 was both positively correlated with hypoxia score and stemness score, the internal relationship and regulatory mechanism between COPZ1 and hypoxia, stemness and immune microenvironment are worthy of further exploration. Meanwhile, COPZ1 expression showed a negative correlation with many antitumor immune signature scores, proinflammatory cytokines and immune activator genes. These results suggested that COPZ1 may be involved in the regulation of the tumor immune microenvironment. However, whether COPZ1 inhibition can sensitize the anti-tumor effect of immune checkpoint inhibitors has not been reported, which is a promising direction for further exploration.

In summary, our study is the first comprehensive analysis of COPZ1 in pan-cancers. Our works revealed the prognosis value of COPZ1 in a variety of cancers and its correlation with stemness score, hypoxia score and tumor immune microenvironment. However, the limitation in our study is that we lack some biological experiments to verify the function of COPZ1, which will be improved in future investigations. Finally, we validated the role of the COPZ1 on the tumor growth and migration in the liver cancer with the biological experiments.

## 5. Conclusions

Our research revealed a relationship between COPZ1 expression and the tumor immune micro-environment, stemness score, and hypoxia score. According to our study’s findings and earlier re-search, COPZ1 appears to be a promising therapeutic target for the treatment of cancer.

## Figures and Tables

**Figure 1 genes-14-00889-f001:**
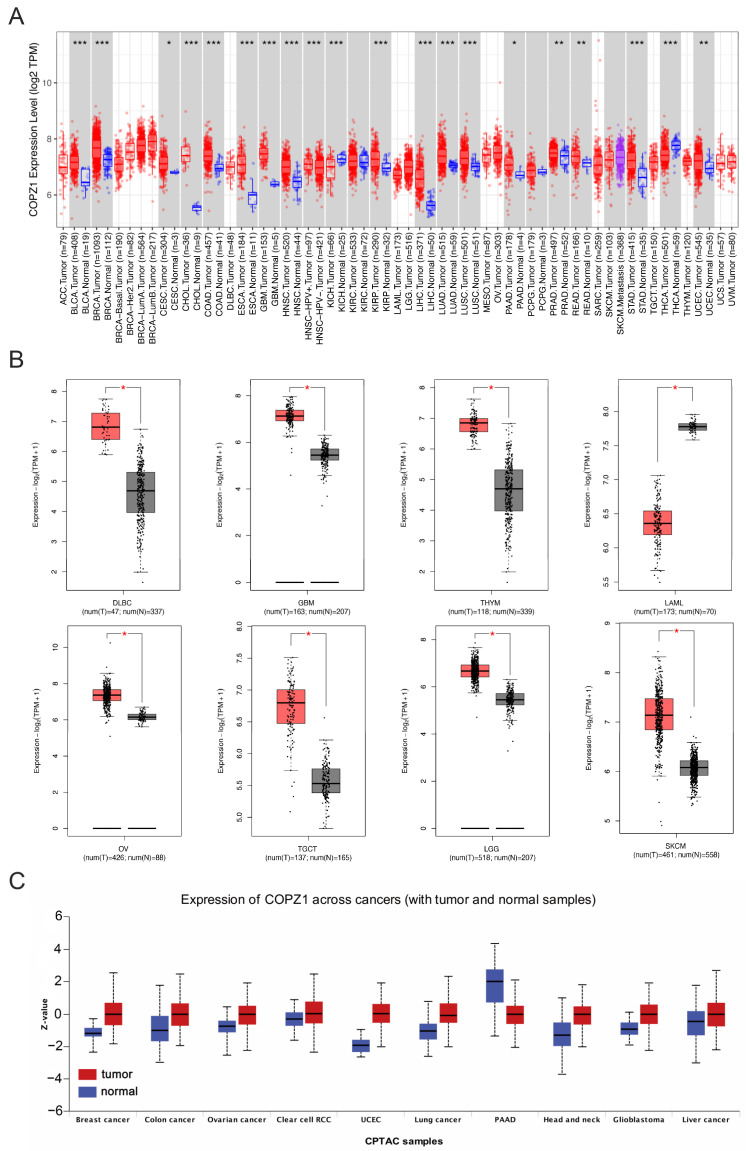
COPZ1 expression profile in pan-cancers. (**A**) The boxplot showed the COPZ1 mRNA expression level in different types of cancer and normal tissues in the TIMER2 database. (**B**) The expression level of COPZ1 in DLBC, GBM, THYM, LAML, OV, TGCT, LGG and SKCM tumor tissues and the corresponding normal tissues of the GTEx database using GEPIA2. (**C**) The protein level of COPZ1 across ten various tumor types and the matching normal tissues the CPTAC database. The red box is tumor tissue, and the blue box is normal tissue. *, *p* < 0.05; **, *p* < 0.01; ***, *p* < 0.001.

**Figure 2 genes-14-00889-f002:**
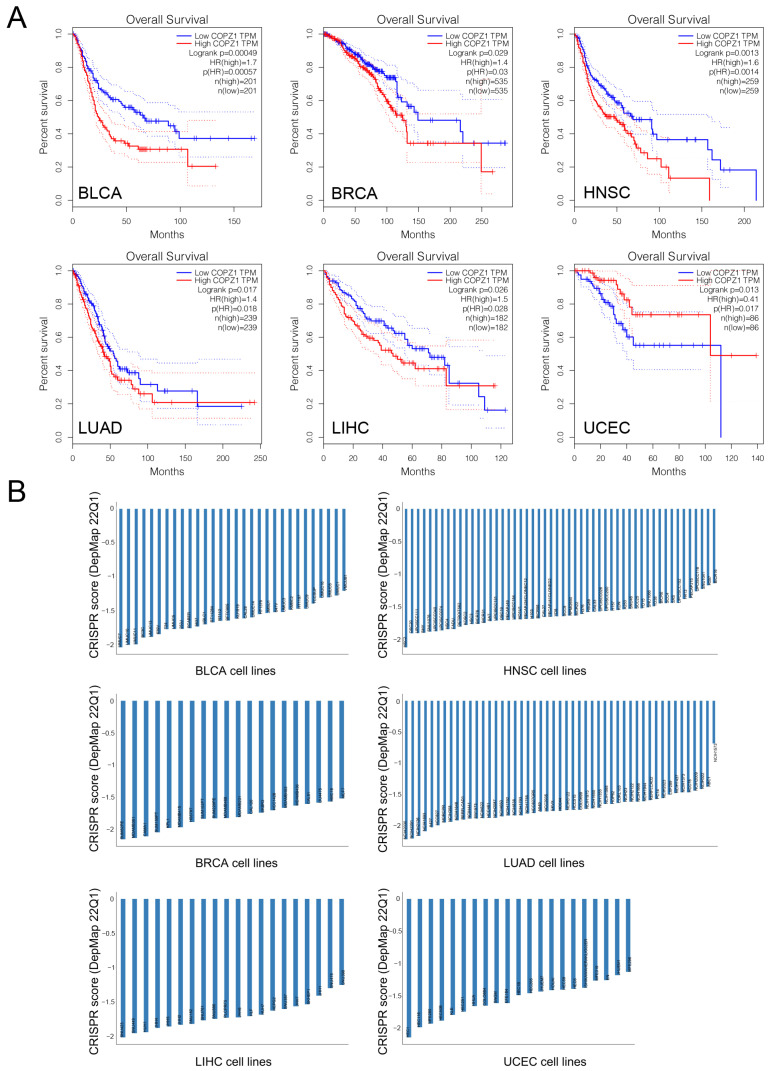
High expression of COPZ1 correlated with poor prognosis. (**A**) Overall survival analysis of COPZ1 in BCLA, BRCA, HNSC, LIHC, LUAD, and UCEC in the TCGA dataset using GEPIA2. (**B**) The Achilles crisper score analysis of the sensitivity of cell lines to COPZ1 knockdown. The score below “0” indicated the cell growth inhibition by COPZ1 knockdown.

**Figure 3 genes-14-00889-f003:**
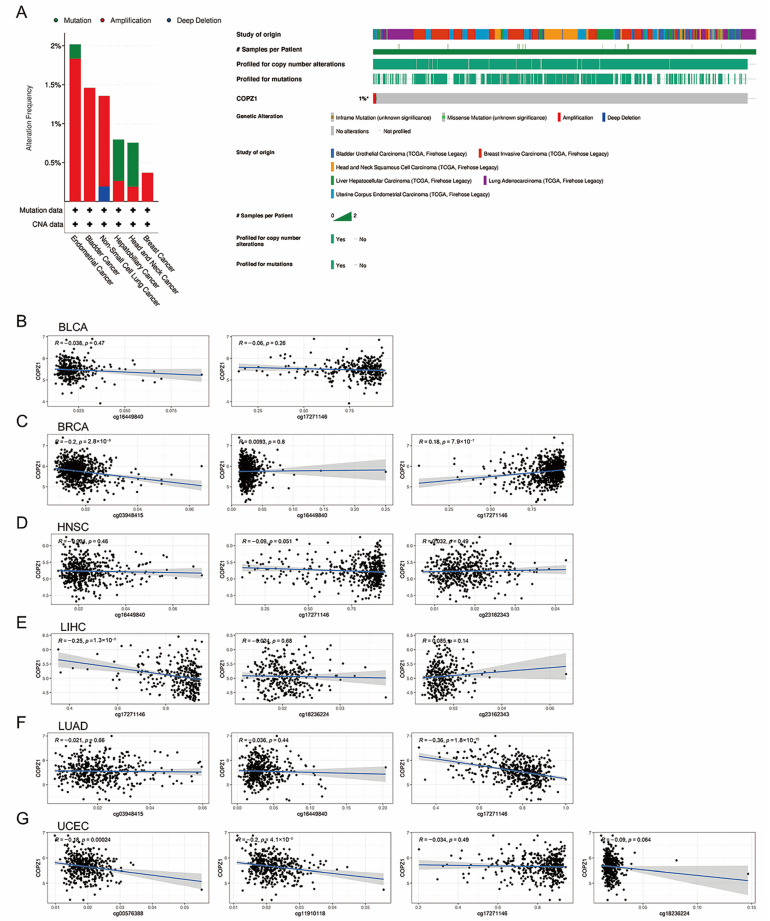
Genetic alterations and DNA methylation analysis of COPZ1 in cancers. (**A**). The alteration frequency and mutation type of COPZ1 were analyzed by the cBioPortal database. (**B**–**G**). The correlation of gene expression and DNA methylation of COPZ1 in six types of cancers.

**Figure 4 genes-14-00889-f004:**
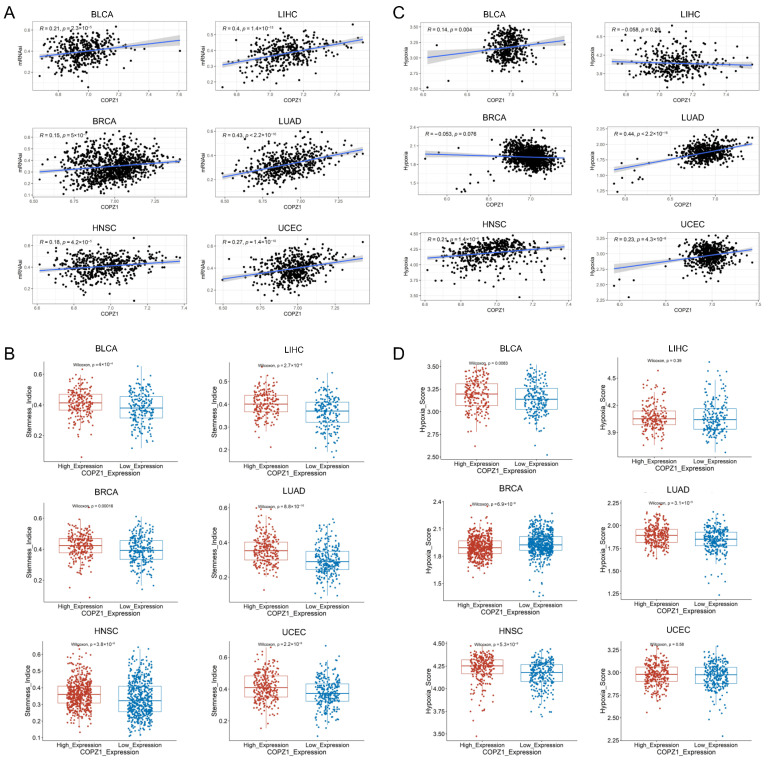
Correlation analysis between COPZ1 expression and stemness score and hypoxia score. (**A**) Correlation between COPZ1 expression and the stemness score. (**B**) The stemness index between high and low COPZ1 expression groups. (**C**) Correlation between COPZ1 expression and the hypoxia score. (**D**) The hypoxia score between high and low COPZ1 expression groups. The red box means high COPZ1 expression group, and the blue box means low COPZ1 expression group.

**Figure 5 genes-14-00889-f005:**
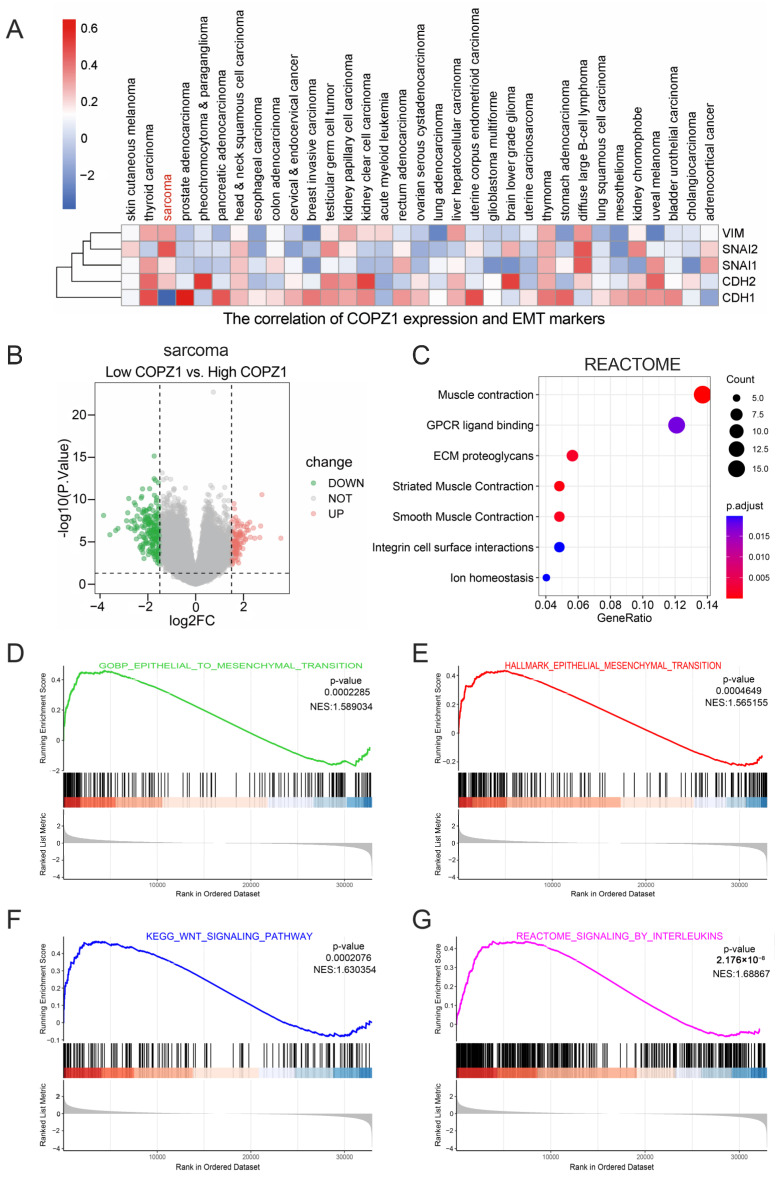
Correlation analysis of the COPZ1 expression and tumor metastasis. (**A**) Heatmap showed the correlation of COPZ1 expression and EMT related markers in 33 types of cancer. (**B**) The volcano plot of the DEGs between high and low COPZ1 expression in SARC. (**C**) The REACTOME enrichment analysis of the DEGs. (**D**–**G**) The GSEA enrichment analysis of the BP, HALLMARK, KEGG and REACTOME in SARC.

**Figure 6 genes-14-00889-f006:**
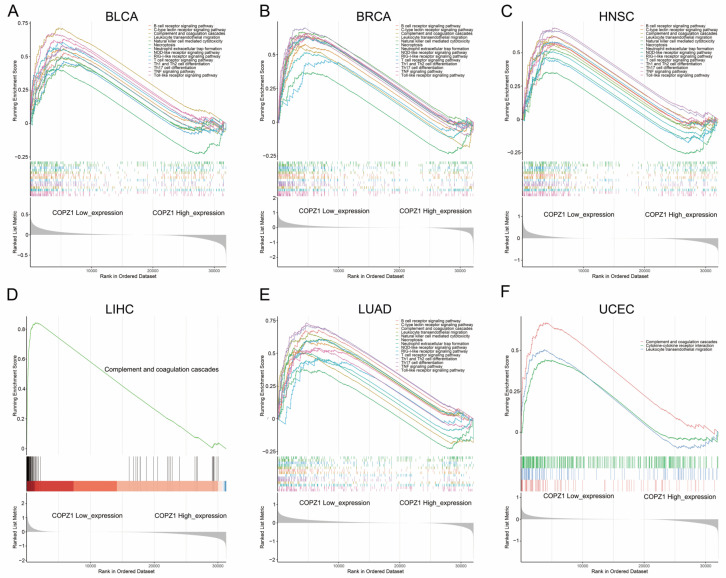
KEGG pathway GSEA analysis of COPZ1 low expression in multiple types of cancer. (**A**) The GSEA analysis showed the pathways activated in the low COPZ1 expression group in the BLCA. (**B**) The GSEA analysis showed the pathways activated in the low COPZ1 expression group in the BRCA. (**C**) The GSEA analysis showed the pathways activated in the low COPZ1 expression group in the HNSC. (**D**) The GSEA analysis showed the pathways activated in the low COPZ1 expression group in the LIHC. (**E**) The GSEA analysis showed the pathways activated in the low COPZ1 expression group in the LUAD. (**F**) The GSEA analysis showed the pathways activated in the low COPZ1 expression group in the UCEC.

**Figure 7 genes-14-00889-f007:**
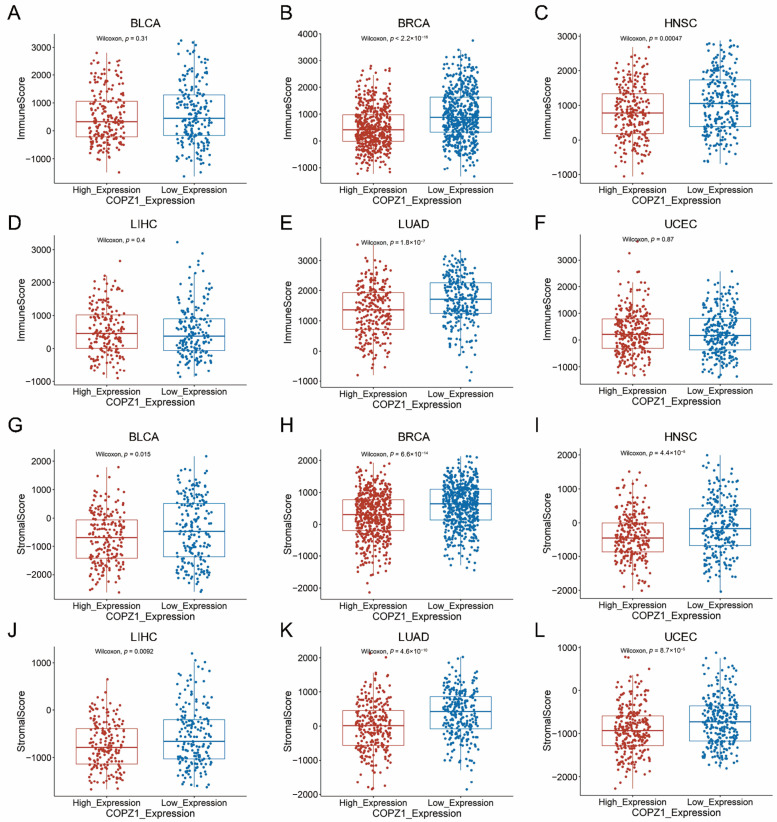
The correlation between COPZ1 expression and immune score and stromal score. (**A**–**F**) The immune score between high and low COPZ1 expression groups in multiple cancers. (**G**–**L**) The stromal score between high and low COPZ1 expression groups in multiple cancers. The red box means high COPZ1 expression group, and the blue box means low COPZ1 expression group.

**Figure 8 genes-14-00889-f008:**
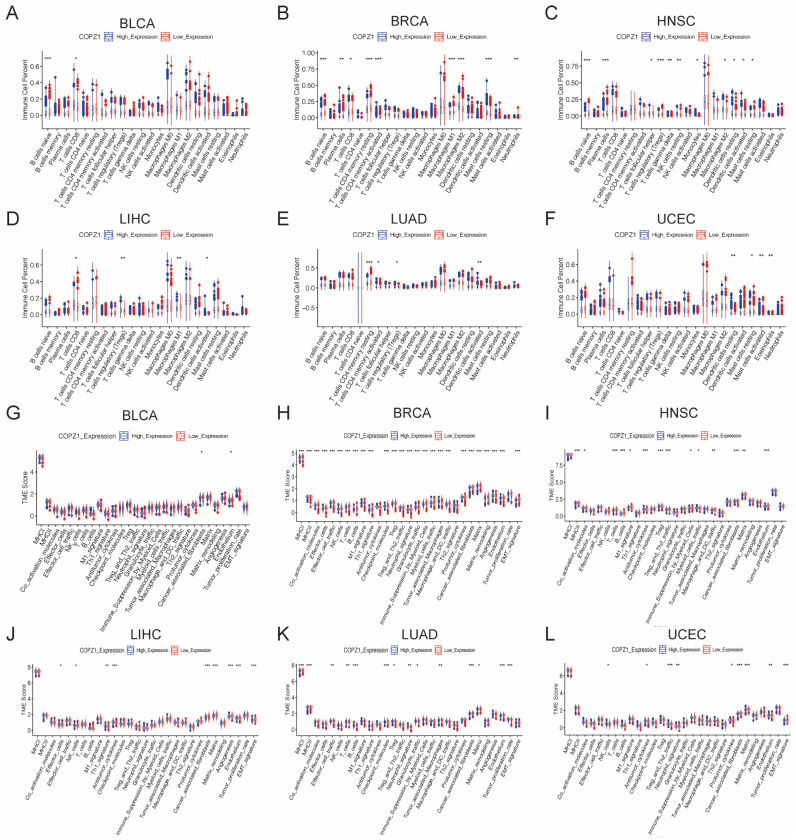
Correlation analysis between COPZ1 expression and tumor-infiltrating immune cells and tumor microenvironment score. (**A**–**F**) I Tumor-infiltrating immune cells between high and low COPZ1 expression groups in six types of cancer. (**G**–**L**) TME score between high and low COPZ1 expression groups in six types of cancer. The red box means high COPZ1 expression group, and the blue box means low COPZ1 expression group. * *p* <0.05, ** *p* <0.01, *** *p* <0.001.

**Figure 9 genes-14-00889-f009:**
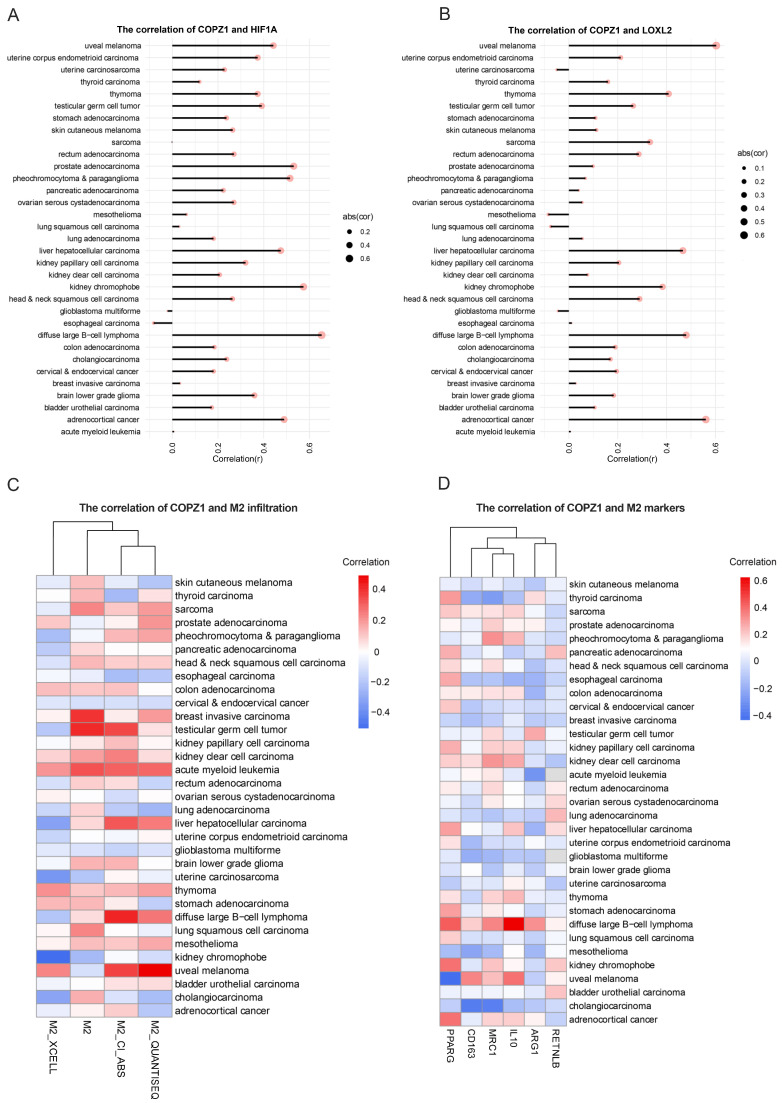
Correlation analysis between COPZ1 expression, hypoxia level and M2 infiltrating level. (**A**) The spearman analysis of COPZ1 expression and HIF1A expression in 33 types of cancer. (**B**) The spearman analysis of COPZ1 expression and LOXL2 expression in 33 types of cancer. (**C**) Heatmap showed the correlation of ZOPC1 expression and the M2 infiltration score generated by XCELL, CIBERSORT, CIBERSORT-ABS and QUANTISEQ in different cancer types. (**D**) Heatmap showed the correlation of ZOPC1 expression and the M2 marker expression.

**Figure 10 genes-14-00889-f010:**
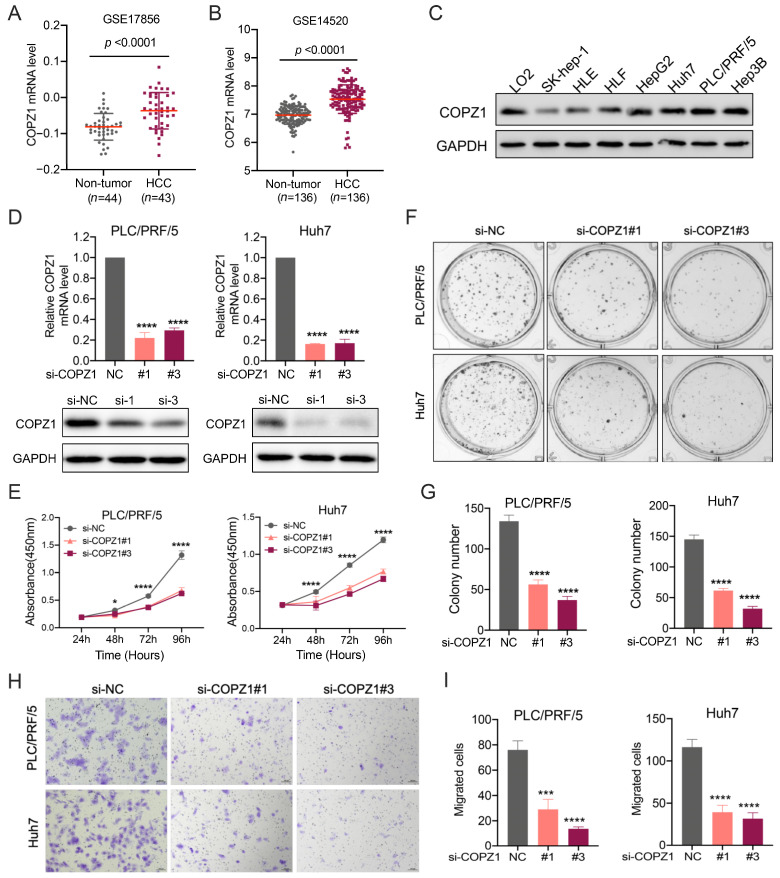
High COPZ1 levels contribute to the proliferation and migration of HCC. (**A**,**B**) The mRNA level of COPZ1 was higher in HCC tissues than the non-tumor tissues in GSE17856 and GSE14520 datasets. (**C**) The protein expression profile of COPZ1 in a panel of HCC cell lines and normal human liver cells. (**D**) Evaluation of the knockdown effect of siRNA targeted for COPZ1 in PLC/PRF/5 and Huh7 cells. (**E**) COPZ1 silence suppresses the cell viability of PLC/PRF/5 and Huh7 cells. (**F**,**G**) The representative images and statistical analysis of colony formation in PLC/PRF/5 and Huh7 cells treated with siRNA targeted for COPZ1 or negative control. (**H**,**I**) The representative images and statistical analysis of the migrated cells in the utter chamber in migration assay in PLC/PRF/5 and Huh7 cells treated with siRNA targeted for COPZ1 or negative control. The data are presented as the mean ± SDs. * *p* < 0.05, *** *p* < 0.001, **** *p* < 0.0001 (Student’s *t*-test).

## Data Availability

The corresponding author or first author will provide data supporting this research study upon reasonable request.

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
