# Peer review of "A Comprehensive Pan-Cancer Analysis of the Regulation and Prognostic Effect of Coat Complex Subunit Zeta 1"

_genes, 2023, doi:10.3390/genes14040889_

Round 1

Reviewer 1 Report

Line 58-62: How are isoforms COPZ1 and COPZ2 different?

You mention that low expression of a COPZ1 is associated with breast, prostate and ovarian cancer (line 57-58). But lines 59-64, discuss how depletion of COPZ1 can lead to thyroid tumor cell death, ER stress, type I IFN pathway activation or autophagy in GBM cells. thus enhancing anti-tumor immunity. How can one explain these different effects i.e. low COPZ1 having both anti-tumor and pro-tumor effects (lines 57-58, 66)? Also how can it be used as a biomarker then?

Lines 149-150: How many normal samples were there (As compared to cancer samples) in GEPIA2 vs TIMER2? 

Were similar trends  (i.e. COPZ1 up or downregulation) observed for the cancers which had datasets in both GEPIA2 and TIMER2 (e.g. BLCA)?

In Fig 1C, use similar nomenclature for the names of the cancers i.e. is breast cancer same as breast invasive carcinoma (BRCA) (Fig 1A,B)?

Line 196: How did authors find “no significant correlation between COPZ1 expression level and COPZ1 mutation among six types of cancer”. What test was performed?

Line 197: Similarly what test was performed to posit that “the CNV level of COPZ1 is significantly different between the tumor tissue and the corresponding normal tissue”?

Line 248: Where was the miRNA data obtained? What test was performed to explore the correlation between significantly downregulated microRNAs and COPZ1 expression?

Line 262: What is mRNAsi score?

Fig. 5 : Explain figure in more details (axes, colours etc.). Also text ineligible in figure.

General comments: Please edit text for grammar (Especially introduction section)

.

Reviewer 2 Report

In this study, authors demonstrated COPZ expression in pan-cancers and its function as a prognostic marker. They analyzed multi-omics data of COPZ and found COPZ1 was related to survival rates. Various factors were correlated with COPZ1 functions, such as CNV, promoter methylation, stemness, and microenvironment. Below are a number of issues that the authors shall address:

1. Authors have analyzed some tumor-related factors which can be the mechanism of COPZ1 in tumor progression. It is better to validate some findings with experiments.

2. In the Figure 1A, higher COPZ1 expression level can be found in SKCM. metastasis than the primary tumor. I wonder whether COPZ1 is related to tumor metastasis and its potential mechanism.

3. In the Figure 7, COPZ1 was highly expressed in the M0 and M2 macrophage. In Line 384-385, Authors also mentioned TAMs could be found in the hypoxia area, which would be converted into M2 macrophages with more immunosuppressive activities. Could authors explain the relationship among COPZ1, hypoxia and TAMs in detail?

Reviewer 3 Report

The authors described a linkage of COPZ1 expression level with overall survival, methylation, pathway enrichment and compared expression level between tumor and normal tissue using public datasets. Also, they found that COPZ1 knockout tumor cells show inhibition of tumor growth. The paper is well written. Hovewer, there is a lack of novelty because differential genes and association with survival have been already described many times for public datasets. Thus, the paper should contain any own experimental or clinical data or new scientific knowledge. Also, it is interesting to find what difference in carcinogenesis leads to inverse trend for COPZ1 in such cancers like pancreatic or LAML than in other cancers. 

Round 2

Reviewer 2 Report

The authors have now addressed my concerns.

Author Response

Thank you very much for your affirmation of our work. We appreciated your time and the constructive feedback. 

Reviewer 3 Report

The authors significantly improved the manuscript. Hovewer, DEGs and pathways  were characterized only for LAML and PAAD to explain the inverse trend, without comparison with other cancers. It is interesting to compare, what DEGs and pathways are:

1)specific for LAML or PAAD but not for other cancers(potentially responsible for the negative trend)

2) common for other cancers and not detected LAML or PAAD as differential ones

Also, GSEA calculates pathway activation or inhibition score(NES) using only gene set enrichment at the top or bottom of the ranked gene list. It does not take into account a role of a gene in the biological process/pathway (a gene may be negative or positive regulator). To solve this problem, a tool with the corresponding function should be used(e.g., OncoboxPD,...) to more accurately characterize the activation or inhibition of molecular pathways.
